# Polycyclic Aromatic Hydrocarbon Risk Assessment and Analytical Methods Using QuEchERS Pretreatment for the Evaluation of Herbal Medicine Ingredients in Korea

**DOI:** 10.3390/foods10092200

**Published:** 2021-09-16

**Authors:** Hee-Jeong Hwang, Sae-Ha Lee, Yong-Yeon Kim, Han-Seung Shin

**Affiliations:** 1Research Institute of Biotechnology and Medical Converged Science, Dongguk University-Seoul, 32, Dongguk-ro, Ilsandong-gu, Goyang-si 10326, Gyeonggi-do, Korea; piatop@hanmail.net; 2Department of Food Science and Biotechnology, Dongguk University-Seoul, 32, Dongguk-ro, Ilsandong-gu, Goyang-si 10326, Gyeonggi-do, Korea; yshwl11@naver.com (S.-H.L.); kimyy613@naver.com (Y.-Y.K.)

**Keywords:** polycyclic aromatic hydrocarbons, herbal medicine, QuEChERS, exposure assessment, risk characterization

## Abstract

Polycyclic aromatic hydrocarbons (PAHs) are carcinogenic and mutagenic compounds that are often formed during the thermal processing of herbal medicine ingredients. In this study, the concentrations of four PAHs (PAH4) in various herbal medicine ingredients were monitored. Further, the QuEChERS method was used to replace conventional pretreatment, a more complex and cumbersome approach. The recovery range of the QuEChERS method ranged between 89.65–118.59%, and the average detection levels of benzo[a]anthracene (BaA), chrysene (CHR), benzo[b]fluoranthene (BbF), and Benzo[a]pyrene (BaP) in 50 herbal medicine ingredients were 0.18, 0.27, 1.13, and 0.17 μg/kg, respectively. The BaP and PAH4 levels in all tested samples were deemed safe according to risk characterization analyses based on European Union and Korean guidelines. Therefore, our findings indicated that the QuEChERS method could be used as an effective alternative to conventional sample pretreatment for the analysis of herbal medicine ingredients.

## 1. Introduction

Although synthetic drugs were proven to be effective treatments for various diseases, interest in natural therapeutic products steadily increased throughout recent years. Particularly, there is a growing interest in herbal medicines that were used for approximately 5000 years as traditional remedies. Nevertheless, there are also growing concerns regarding the carcinogenic compounds that are unintentionally generated during the heat treatment process of herbal medicine ingredients, particularly polycyclic aromatic hydrocarbons (PAHs). PAHs can be generated from the incomplete combustion of organic matter or can be produced when organic sediments are chemically transformed into fossil fuels such as oil and coal [1]. PAHs are formed in two steps: pyrolysis and pyrosynthesis. Organic materials become partially decomposed and divided into unstable fractions when they reach a high temperature (pyrolysis). These fractions are then recombined to become extremely reactive radicals, which are stable PAHs (pyrosynthesis) [2]. PAHs are formed during the combustion process of carbonaceous materials at high temperatures [3]. Applying high temperatures to herbal medicine ingredients during the manufacturing process can thus lead to PAH generation. Also, relocation of PAHs to the raw material in environments such as soil, air, and so on can be involved in the detection of PAHs.

PAHs pose a severe risk to human and environmental health. Furthermore, these compounds are highly pervasive and persistent, and thus, remain in the environment for long periods by interacting with particles in the soil, sediment, and air, resulting in severe pollution. PAHs with two or more aromatic rings in a linear, cluster, or angular arrangement were linked to tumorigenesis in humans [4]. Therefore, the International Agency for Research on Cancer (IARC) classified PAHs as a working group to evaluate their carcinogenicity to humans [5]. Benzo[a]pyrene (BaP) is considered a Group1 compound (carcinogenic to humans), whereas benzo[a]anthracene (BaA), chrysene (CHR), and benzo[b]fluoranthene (BbF) are classified as Group2B (possibly carcinogenic to humans). In December 2009, the Korea Food & Drug Administration (KFDA) established the benzopyrene standard for all herbal medicines except for mineral herbal medicines to be less than 5 μg/kg, and they also announced a test method [6].

PAH analyses are conducted in many areas, particularly the food industry; however, sample pretreatment is time-consuming and highly inconvenient. Determination of PAH in herbs, tea, and edible oils are often carried out using gas chromatography (GC) and high-performance liquid chromatography (HPLC) [7,8,9]. To address these methodological challenges, the quick, easy, cheap, effective, rugged, and safe (QuEChERS) method was developed [10]. This approach has become one of the most commonly used methods for the analysis of multiclass pesticide residues in vegetables and fruits, and more recently was used to detect traces of contaminants such as PAHs [8,11,12]. This study sought to monitor four PAHs (PAH4) contents in herbal medicine ingredients using the QuEChERS method coupled with HPLC-FLD and evaluate the efficiency of this approach by comparing it with that of the conventional pretreatment method coupled with GC/MS. Further, risk assessment analyses were conducted by calculating toxicity equivalency (TEQ), the daily average intake (DAI), and margin of exposure (MOE) to evaluate cancer risks in the Korean population.

## 2. Materials and Methods

### 2.1. Chemicals and Materials

The PAH4 standards benzo[a]pyrene (BaP, CAS No. 50-32-8), benzo[b]fluoranthene (BbF, CAS No. 205-99-2), chrysene (CHR, CAS No. 218-01-9), and benzo[a]anthracene (BaA, CAS No. 56-55-3) and internal standards (IS) 3-methylcholanthrene (CAS No. 56-49-5), chrysene-d12 (CAS No. 1719-03-5), and benzo[a]pyrene-d12 (CAS No. 63466-71-7) were obtained from Supelco, Inc. (Bellefonte, PA, USA). All materials and chemicals were of GC and HPLC analytical grade. Acetonitrile (ACN) (CAS No. 75-05-8), dichloromethane (DCM) (CAS No. 75-09-2), methanol (CAS No. 67-56-1), and hexane (CAS No. 110-54-3) were obtained from Burdick & Jackson (Muskegon, MI, USA). Sodium sulfate anhydrous (Na_2_SO_4_) (CAS No. 7757-82-6) 99% *w*/*w* purity; (Samchun Co., Ltd., Seoul, Korea) was used for dehydration. Distilled water (CAS No. 7732-18-5) and N,N-dimethylformamide (CAS No. 68-12-2) were prepared using a Milli-Q water purification system (Millipore, Billerica, MA, USA). Kovax syringes (3 mL) were obtained from Korea Vaccine Co., Ltd. (Gyeonggi-do, Korea). Polytetrafluoroethylene (PTFE) membrane filters (0.45 mm) were purchased from Advantec Co., Ltd. (Chiyoda City, Japan). Sep-Pak Florisil cartridges (Waters Corp., Milford, MA, USA) were utilized for solid-phase extraction (SPE).

### 2.2. Sample Preparation for PAH4 Evaluation

From January to September 2019, 50 types of herbal medicine products were provided by the Ministry of Food and Drug Safety (MFDS) in the Republic of Korea to evaluate PAH4 contents. The samples included milk-thistle fruit, fresh bilberry fruit, ginkgo leaf, angelica gigas root, grape seed, chaenomelis fructus, saposhnikovia root, dipsaci radix, acanthopanax root bark, archyranthes root, clematis radix, cinnamon bark, gentianae macrophyllae radix, cnidium rhizome, gastrodia rhizome, safflower, *Vitis vinifera* seed, and ivy leaf, among others. The samples were chopped into small-sized pieces and stored at −18 °C prior to the analysis.

### 2.3. Extraction and Clean-Up

The homogenized herbal medicinal samples were weighted (5 g) in round bottom flasks containing 100 mL of water for ultrasonic extraction (60 min). After this water ultrasonic extraction step, hexane (100 mL) was added for an additional ultrasonic extraction (60 min) and spiked with 1.0 mL of 50 μg/kg IS (3-methylcholanthrene, BaP-d12, and CHR-d12). For liquid-liquid extraction, the mixtures were centrifuged at 3200 g for 10 min, and the upper hexane layer was transferred to a separatory funnel. After liquid-liquid extraction, 50 mL of N,N-dimethylformamide/water (9:1; *v*/*v*) was added to the hexane layer of the separatory funnel, after which the N,N-dimethylformamide/water (9:1; *v*/*v*) layer was transferred to another separatory funnel after shaking; this extraction procedure was repeated three times. Next, 100 mL of 1% sodium sulfate solution and 50 mL of hexane were added to the separatory funnel, after which the separated hexane layer was transferred to another separatory funnel; thirty-five mL of hexane was added to the separatory funnel to extract with shaking. This hexane layer was recovered, and the extraction procedure was repeated twice. The hexane layer was then washed with 50 mL of water and conjugated via dehydrating filtration in 30 g of Na_2_SO_4_ in a round-bottomed flask. The extract was concentrated to 2 mL in a water bath at 45 °C under reduced pressure (70 kPa) in a vacuum rotary evaporator.

Purification was carried out after the extraction process. Firstly, the concentrated extract was separated from the impurities via solid-phase extraction using Sep-Pak Florisil cartridges (Waters Corp., Milford, MA, USA) preconditioned with 10 mL of DCM and 20 mL of hexane eluted at a rate of 2 drops per second. Next, the extract was eluted from the SPE cartridge with 15 mL of DCM and 5 mL of hexane:DCM (3:1; *v*/*v*). This eluted solution was dried in a heating block at 35 °C under a constant nitrogen gas flow. The residues were dissolved with 1 mL of ACN by vortexing. The testing solution was filtered with a 0.45-µm PTFE membrane syringe filter and transferred to amber vials with a screw cap for HPLC-FLD and GC-MS analyses.

### 2.4. Preprocessing of Samples Using QuEChERS

The QuEChERS method entails an extraction step and a clean-up step. For the extraction step, we employed the QuEChERS Performance Standards Kit (Restek, GmbH Schaberweg 23, Bad Homburg, Germany; Cat. No. #25847) containing 4 g MgSO_4_ for dehydration and 1 g NaCl to maintain the pH. For preprocessing, the samples were homogenized to increase their surface area. The homogenized 1 g of samples and 10 mL of distilled water were transferred to a 50 mL conical tube (Supermax Corporation Berhad, Kuala Lumpur, Malaysia; Cat. No. #S20050). After 1 h, 10 mL of hexane:acetone (1:1; *v*/*v*) was added and the mixture was vortexed for 10 min. After adding the QuEChERS salts, the samples were centrifuged at 3000 g for 5 min. For the clean-up step, Resprep silica SPE cartridges (Restek, GmbH Schaberweg 23, Bad Homburg, Germany; Cat. No. #24036) were used to extract hydrophilic analytes from nonpolar matrices. The silica cartridge was rinsed with 3 mL of methanol and 3 mL of acetone under high vacuum conditions with 3 mL of hexane:methylene chloride (1:1; *v*/*v*) and 6 mL of hexane at a 1 drop per second rate. The supernatant in the conical tube was then transferred to a conditioned silica cartridge, followed by 5 mL of hexane:methylene chloride (85:15; *v*/*v*) solution. This extract was evaporated using a gentle stream of nitrogen gas in a heating block at 50 °C. Finally, 1 mL of acetonitrile was used to dissolve the extract for HPLC-FLD analyses.

### 2.5. GC-MS Analysis of PAH4

PAH4 in the samples processed via the conventional pretreatment method was analyzed using a GC-MS system (Agilent Technologies 7820A/5975 MSD GC-MS system, Santa Clara, CA, USA) equipped with an HP-5MS UI capillary column (0.25 mm × 30 m i.d., 0.25 µm particle size). Helium gas (99.99%) was used as a carrier at a constant flow rate of 1.0 mL/min. The oven temperature was set to 80 °C, held for 1 min, increased to 245 °C at a 10 °C/min rate, then to 290 °C at a 10 °C/min rate, and finally held postrun at 310 °C for 5 min. The sample solution was injected in splitless mode at 310 °C, and the injection volume was 1.0 mL. The quadrupole temperature was 150 °C and the MS source temperature was 250 °C. Each of the IS and the PAH4 had two qualifier ions and one target ion (underlined). The selected ions were 228, 226, and 229 for CHR and BaA; 252, 250, and 253 for BaP and BbF; 240, 236, and 241 for CHR-d12; and 264, 263, and 265 for BaP-d12. PAHs were identified via extracted ion chromatogram, and the molecular mass was determined using the GC-MS program.

### 2.6. HPLC-FLD Analysis of PAH4

PAH4 in the samples treated via the QuEChERS pretreatment method were analyzed using an HPLC-FLD analysis instrument (Dionex U3000 HPLC coupled with a fluorescence detector, Thermoherbal Medicineer, Sunnyvale, CA, USA) equipped with a ZORBAX eclipse C18 plus column (4.6 mm id × 250 mm × 5 µm, Agilent, Santa Clara, CA, USA). The injection volume was 10 µm at a 1.0 mL/min flow rate. Acetonitrile and water were used as the mobile phase at a 65:35 (acetonitrile:water; %) ratio from 0 to 20 min and 70:30 (%) from 20 to 60 min. The excitation and emission wavelengths were 245/390 nm from 0 to 30 min and 294/404 nm from 30 to 60 min. Separation was conducted with the following gradient program: 35% B for 20 min and 30% B for the last 30 min.

### 2.7. Identification and Quantification of PAH4

PAH4 were identified by comparing their retention times and those of standards. For validation, 5 concentrations of the PAH4 solutions (3, 5, 10, 20, and 40 μg/kg) containing 50 μg/kg of the IS mixture were evaluated.

### 2.8. Method Validation for Analytical Quality Assurance

All analytical methods were validated for limit of detection (LOD), limit of quantification (LOQ), linearity, precision (%), and recovery (%) on herbal medicine samples. Calibration curves were created from 50 μg/kg of IS mixture (3-methylcholanthrene; CHR-d12 and BaP-d12) and PAH4 standard mixtures at 3, 5, 10, 20, and 40 μg/kg.

### 2.9. Application of TEQ Concentration

BaP is the most widely recognized PAH and Class1 carcinogenic compound. Human exposure was evaluated by analyzing herbal medicine products spiked with 3-methylcholanthrene, CHR-d12, and BaP-d12. This study was conducted to estimate the PAH exposure levels of the entire Korean population.

The relative toxicity coefficient of congener *i* based on BaP cancer potency (Toxic equivalency factor, TEFs) was estimated as BaP equivalents, considering that each PAH has a different toxicity level. BaP concentration conversion was conducted using TEQ by multiplying each PAH by their respective TEF:(1)$$\mathrm{TEQ}=\sum_{i=1}^{n}\left[Ci\right]\;\times \mathrm{TEF}i,$$
where *Ci* is the concentration of each PAH congener in herbal medicine products, and TEF*i* is the relative toxicity coefficient of congener *i* based on BaP cancer potency corresponding to BaP.

TEQ data were acquired from the *Ci* of PAHs among many herbal medicine products based on analytical methods and the TEF*i* proposed by [13] (i.e., the latest study on herbal medicine products). Given the differences in sample numbers for each herbal medicine product, the average concentrations of PAH4 in herbal medicine products were not determined.

### 2.10. Exposure Assessment

Dietary intake is the main route of PAH exposure in humans. Therefore, PAH exposure in adults and children was estimated based on oral consumption of leading herbal medicine products.

Daily exposure to PAH4 was calculated based on total lifetime intake. By applying the BaP equivalent concentrations, PAH4 concentrations, and the daily consumption of herbal medicine products, the DAI was calculated as follows:(2)$$\mathrm{DAI}(\mu g/\mathrm{kg}/\mathrm{day})=\sum_{i=1}^{n}\frac{Ci\times IRi\times \mathrm{ED}}{\mathrm{BW}\times \mathrm{AT}},\;$$
where *Ci* is the respective TEQ of the PAH4 in herbal medicine *i* (μg/kg); *IRi* is the average daily intake of herbal medicine *i* according to the National Health and Nutrition Survey (ingestion rate, 0.0061 g/day); AT is the average life expectancy (80.4 years); BW is the body weight by age group (64 kg); ED is the exposure period (45 years). The AT and BW values were collected from the 2018 statistics data of Korea [13] and the 2018 National Health Screening Statistical Yearbook of Korea.

### 2.11. Exposure Assessment

To assess risk, the MOE was calculated based on the benchmark dose lower confidence limit (BMDL) (mg/kg·BW/day) and the dietary exposure (mg/kg·BW/day). *CBi* is the concentration of BaP (mg/kg).
(3)$$\mathrm{Dietary}\;\mathrm{exposure}=\sum_{i=1}^{n}\frac{CBi\times IRi}{\mathrm{BW}},\;$$
(4)$$\mathrm{MOE}=\frac{\mathrm{BMDL}}{\mathrm{Dietary}\;\mathrm{exposure}},\;$$

The Committee on Carcinogenicity of Chemicals in herbal medicine, Consumer Products, and the Environment classified the risk of MOE values <10,000 as “possible concern,” 10,000–1,000,000 as “low concern,” >100,000 as “negligible concern with action minimizing future exposure,” and >1,000,000 as “negligible concern” [14]. In other words, the level of concern decreases as the MOE value increases.

Excessive cancer risk was estimated based on the cancer risk of BaP and the DAI values as follows:(5)Excessive cancer risk=cancer risk of BaP×DAI, 
where BaP’s cancer potency is 7.3 (mg/kg/day)^−1^ based on the U.S. Environmental Protection Agency’s integrated risk information system. Excessive cancer risk values >10^−4^ are considered a “serious risk,” those between 10^−6^–10^−4^ are considered a “potential risk,” and those <10^−6^ are considered “safe and acceptable.”

### 2.12. Statistical Analyses

All analyses were conducted in triplicate and the data were expressed as mean ± standard deviation (SD) using Microsoft Excel (version 2016, Microsoft, Redmond, WA, USA).

## 3. Results

### 3.1. Method Validation and Confirmation of PAH4 in Herbal Medicine

The GC-MS and HPLC chromatograms of PAH4 standards (A), PAH4 with spiked sample (B), two internal standards with a blank sample (C), and chromatograms of PAHs for each sample (D) are presented in Appendix A. A calibration curve was generated based on five different standard mixture concentrations (3, 5, 10, 20, 40 µg/kg) for the validation of the conventional sample pretreatment method via GC-MS. The linearity, LOD, and LOQ of the PAH4 are summarized in Table 1. All correlation coefficients (R^2^) values for PAH4 exceeded 0.99, and the LOD and LOQ values were 0.08–0.15 µg/kg and 0.24–0.45 µg/kg, respectively. Likewise, regarding the validation of the QuEChERS pretreatment method coupled with HPLC-FLD, the correlation coefficient (R^2^) exceeded 0.99 for PAH4 at all concentrations. The LOD and LOQ values ranged from 0.08–0.17 µg/kg and from 0.25–0.51 µg/kg, respectively.

Further, as shown in Table 2, recovery and precision were evaluated by repeating all five concentrations three times within a day (intraday) and once again for two days (interday). When using the conventional sample treatment method, the recovery and precision values for the intraday experiment were 93.19–117.26% and 0.04–7.84%, respectively. For the interday experiment, the recovery values varied from 91.51–119.51%, whereas precision varied from 0.11–5.71%. The intraday recovery and precision values for the QuEChERS method were 89.65–118.59% and 0.17–2.15%, respectively. Interday recovery and precision varied from 95.31–117.87% and from 0.09–9.38%, respectively.

### 3.2. Comparison of the Two Different Sample Pretreatment Methods for the Analysis of Herbal Medicine Ingredients

PAH4 in herbal medicine ingredients was monitored with both the conventional and QuEChERS methods after each of the parameters for method validation was evaluated. An established validation method was employed for the simultaneous determination of PAH4 in 50 herbal medicine ingredients. Table 3 and Table 4 summarize the PAH4 concentrations in samples obtained using two different sample pretreatment methods: (1) conventional sample pretreatment method using GC/MS, and (2) QuEChERS coupled with HPLC-FLD. When using the conventional pretreatment method, the BaA, CHR, BbF, and BaP detection ranges were 0–0.40 μg/kg, 0–0.46 μg/kg, 0–1.84 μg/kg, and 0–0.27 μg/kg, respectively. The QuEChERS method rendered BaA, CHR, BbF, and BaP detection ranges of 0–0.63 μg/kg, 0–0.82 μg/kg, 0.26–3.23 μg/kg, and 0–0.48 μg/kg, respectively. When using the conventional pretreatment method, BaA, CHR, BbF, and BaP were detected in 41, 41, 30, and 21 samples out of 50, whereas this compound was detected in 21, 41, 49, and 41 samples out of 50 when the QuEChERS method was used. Regardless of the type of pretreatment method, sample 32 showed the highest concentration of BaA, CHR, BbF, and BaP. Further, the PAH4 value obtained by combining the concentrations of the four substances was 2.97 μg/kg when using the conventional pretreatment method and 5.16 μg/kg when using the QuEChERS method.

### 3.3. Exposure Assessment

The average TEQ values for the BaP and PAH4 of herbal medicine ingredients were determined based on the TEF values (0.10, 0.01, 0.10, and 1.00 for BaA, CHR, BbF, and BaP, respectively) proposed by [13], as shown in Table 5 and Table 6. The TEQ values obtained using the conventional pretreatment method were 0.00–0.27 µg/kg for BaP and 0.00–0.50 µg/kg for PAH4. When using the QuEChERS method, the TEQ values for BaP and PAH4 were 0.00–0.48 µg/kg and 0.03–0.88 µg/kg, respectively.

The DAI values of the samples were also calculated using each of the derived TEQ values (Table 7). As indicated in Equation (2) and mentioned in Section 2.10, DAI values are proportional to the TEQ value. The values obtained using the conventional pretreatment method were 0–1.42 × 10^−8^ µg/kg/day for BaP and 0–2.64 × 10^−8^ µg/kg/day for PAH4. When using the QuEChERS method, the values were 0–2.57 × 10^−8^ µg/kg/day for BaP and 1.38 × 10^−9^–4.67 × 10^−8^ µg/kg/day for PAH4.

### 3.4. Risk Characterization

Based on the data calculated through the exposure assessment, population-wide MOE values for the herbal medicine ingredients were obtained using the BMDL (100 µg/kg·BW/day) and the dietary exposure. The MOE values for the conventional and QuEChERS methods were 1.30 × 10^10^ and 6.26 × 10^9^, respectively. These two values were >1,000,000, thus indicating a “negligible concern.” The excessive cancer risk values of PAH4 for the total population obtained using the conventional and QuEChERS pretreatment methods were 5.61 × 10^−11^ and 1.17 × 10^−10^, respectively, and therefore the PAH4 concentration level of the herbal medicine ingredient was deemed “safe and acceptable.”

## 4. Discussion

To improve the recovery of the QuEChERS method, selecting an appropriate extraction solvent with a similar polarity to that of the 4 PAHs is key. Most related studies reported that acetonitrile, acetone, and ethyl acetate were suitable extraction solvents because they had an appropriate polarity for most compounds and rendered good recovery rates [15]. In some cases, 1% acetic acid was added to the acetonitrile solution to improve recovery [16,17]. However, more recent studies reported that using hexane:acetone (1:1) as the extraction solvent instead of acetonitrile results in higher PAH recovery rates, and is therefore widely used for the determination of pesticides as an extraction solvent in the QuEChERS method [18]. Acetone is mixed with hexane to induce a distinct separation from the water phase, as hexane dissolves nonpolar molecules [15]. Thus, this solvent mixture is mainly used for the extraction of nonpolar compounds and is therefore suitable for the determination of the four PAHs examined herein, which are nonpolar molecules [19]. Furthermore, a 1 h hydration step is implemented before applying the extraction solvent to the sample to facilitate proper partitioning [15,20]. For the silica SPE clean-up step, larger molecular weight compounds required a stronger solvent to elute them from the silica. Therefore, 15% methylene chloride in hexane was employed for the 4 PAHs, which are considered mid-sized molecules. In this sense, the choice of solvent for 4PAH analysis is an important factor to optimize recovery rates. According to the European Commission (EC) Regulation No. 836/2011, the criteria for analyzing PAH4 (BaA, CHR, BbF, and BaP) are LOD ≤ 0.30 µg/kg, LOQ ≤ 0.90 µg/kg, and recovery values of 50–120% [21]. [22] validated a method for PAH determination in wastewater and sediments and achieved an R^2^ > 0.99, 0.02–0.51 µg/kg LOD, 0.05–1.71 µg/kg LOQ, and 80–104% recovery. [23] also validated a method to determine PAH8 in ready-to-eat food products. The correlation coefficients (R^2^) were higher than 0.99, the LOD and LOQ were 0.12–0.19 µg/kg and 0.36–0.57 µg/kg, respectively, and the recovery and precision were 82.4–113.6% and 0.6–12.4% for interday analysis and 81.2–113.7% and 1.7–13.1% for intraday analysis, respectively. Therefore, the validation parameter values indicated that GC/MS and HPLC-FLD were suitable for the determination of PAH4 in herbal medicine ingredients.

The main purpose of this study was to determine whether the QuEChERS coupled with HPLC-FLD could replace the conventional pretreatment method coupled with GC/MS for the determination of PAH4 in herbal medicine ingredients. No substantial differences in sample concentration and recoveries were observed between the two methods. Nevertheless, the QuEChERS method was better suited for the detection of BaP and PAH4 concentrations. Few studies employed this method for the detection of PAHs in herbal medicine ingredients; however, some studies used this approach to detect residual pesticides in herbs or plants. Ref. [24] developed a QuEChERS-based method for the detection and quantification of pesticides in herbs and achieved recovery rates of 78.4–119.2% and relative standard deviations below 9.5%. Using this method, the authors determined that each sample contained at least one of the examined pesticides. Other studies also applied QuEChERS to dried herbs and plants to detect pesticides and achieved 70%–120% recovery rates [25,26]. The authors thus concluded that this method could be used to monitor pesticides in herbs and plants. Due to the lack of information regarding the application of QuEChERS methods for the assessment of herbal medicine ingredients, assessing the viability of this approach to substitute the conventional pretreatment method was critical. Our results indicated that both the conventional and QuEChERS methods rendered similar recovery ranges and PAH4 detection concentrations. Therefore, we concluded that the QuEChERS method is an effective approach that could potentially replace the conventional PAH4 pretreatment method for the analysis of herbal medicine ingredients used in Korea. However, herbal medicine ingredients contain large amounts of coextractives, and therefore, further studies are required to assess the performance of the modified QuEChERS pretreatment method. Sadowska–Rociek et al. [8] applied the QuEChERS method to analyze black, green, red, and white tea. The authors mentioned that some modifications had to be made to ensure the successful determination of PAHs, as tea contains a variety of interfering substances such as caffeine, polyphenols, and chlorophyll, all of which impede accurate PAH determination.

The specific structures of the herbal medicine ingredients used in this study (i.e., roots, stems, flowers, fruits, seeds, leaves, or bark) were the same as those reported by [27]. PAH4 are often formed in herbal medicine ingredients during thermal processes such as roasting, smoking, or drying. In a study by [9], PAHs ranged from 6.5 to 1112.1 ng/g in tea products and crude herbal medicine ingredients. Further, Ref. [28] reported PAHs of 0.2–11.9 µg/kg in Chinese medicinal herbs. In Korea, the BaP and PAH4 concentration limits in herbal medicine ingredients are 5.0 and 10.0 µg/kg, respectively. All 50 samples evaluated in this study exhibited BaP and PAH4 levels that were below the aforementioned guidelines when using both pretreatment methods and were thus considered safe. Similarly, [29] analyzed 93 herbal pills in Seoul, Korea, and reported that the PAH concentration of all of the samples was below 10 µg/kg. Therefore, the PAH levels in herbal medicine ingredients used in Korea were deemed safe. Based on exposure assessment and cancer risk characterization, our study confirmed that the levels of PAH4 in herbal medicine ingredients were within safe and acceptable limits. Our findings were consistent with those reported by other studies [29,30], which confirmed that tea leaves and herbal pills, respectively, contained safe PAH levels.

## 5. Conclusions

This study evaluated whether the QuEChERS pretreatment method coupled with HPLC-FLD could replace the conventional pretreatment method coupled with GC/MS for the determination of PAH4 (BaA, CHR, BbF, and BaP) in herbal medicine ingredients. Both methods exhibited largely similar BaP and PAH4 detection performances. Further, the results of both pretreatment methods were used to evaluate the health risks associated with BaP and PAH4 in 50 herbal medicine ingredients. Based on European Union and Koreas toxicity guidelines, the BaP and PAH4 concentrations in all samples were deemed safe. Through exposure assessment and cancer risk characterization, the PAH4 levels in various herbal medicine ingredients were found to be within safe and acceptable limits. Taken together, our findings confirm that the QuEChERS method could effectively replace the conventional pretreatment method, thus providing a more practical means for the detection of PAHs in herbal medicine ingredients.

## Figures and Tables

**Table 1 foods-10-02200-t001:** Comparison of typical sample treatment and QuEChERS in regard with linearity with equation of calibration, limit of detection (LOD), and limit of quantification (LOQ) of PAH4 in herbal medicine.

PAHs	Tretment	Linearity ^(1)^	R^2^	LOD (µg/kg) ^(2)^	LOQ (µg/kg) ^(3)^
BaA	Typical	y = 0.0245x − 0.0054	0.999	0.14	0.43
	QuEChERS	y = 0.0115x + 0.0075	0.999	0.12	0.37
CHR	Typical	y = 0.018x + 0.0026	0.999	0.14	0.42
	QuEChERS	y = 0.0104x + 0.0009	0.999	0.17	0.51
BbF	Typical	y = 0.00379x + 0.0088	0.998	0.15	0.45
	QuEChERS	y = 0.0027x + 0.0002	0.999	0.14	0.41
BaP	Typical	y = 0.0273x + 0.0075	0.999	0.08	0.24
	QuEChERS	y = 0.0183x − 0.0003	0.999	0.08	0.25

^(1)^ Numbers express the mean values (*n* = 3). ^(2)^ Set up in a signal-to-noise ratio (*S/N*) = 3.3. ^(3)^ Set up in a signal-to-noise ratio (S/N) = 10.

**Table 2 foods-10-02200-t002:** Recovery and precision comparison of typical sample treatment and QuEChERS for PAH4 in herbal medicine ingredients.

PAHs	Concentration (µg/kg)	Intraday (*n* = 3)				Interday (*n* = 3)			
		Recovery (%)		Precision (%)		Recovery (%)		Precision (%)	
		Typ	QuE	Typ	QuE	Typ	QuE	Typ	QuE
BaA	3	110.93	89.65	0.61	0.45	101.99	104.67	1.34	9.38
	5	113.38	111.22	0.25	0.82	119.24	117.81	2.83	0.69
	10	116.83	111.17	0.4	0.85	119.51	111.74	0.87	0.65
	20	110.69	102.09	0.46	0.44	116.72	108.74	0.11	0.5
	40	112.68	109.68	0.37	0.28	113.1	110.08	1.05	0.59
CHR	3	109.79	102.7	1.11	2.15	109.88	98.34	1.23	3.63
	5	104.78	118.59	1.09	0.46	106.15	117.87	0.84	0.09
	10	105.98	109.8	0.04	0.91	106.35	110.96	0.33	0.51
	20	104.65	103.57	0.1	0.69	105.22	107.58	0.66	1.38
	40	101.73	109.47	0.47	0.46	102.21	108.91	1.08	0.74
BbF	3	101.2	99.69	7.84	0.19	103.73	103.63	5.71	6.9
	5	117.26	112.57	1.47	1.41	114.16	114.54	3.00	4.05
	10	107.69	103.5	0.64	0.84	113.6	103.77	5.06	3.48
	20	106.83	105.45	0.38	0.32	108.42	105.4	3.32	0.87
	40	95.42	104.12	0.69	0.58	93.75	104.83	2.97	0.65
BaP	3	111.33	99.44	6.67	1.01	109.58	95.31	1.85	1.00
	5	112.57	100.21	1.31	0.27	109.20	101.06	5.52	1.58
	10	98.9	103.79	1.5	0.51	100.14	106.45	0.54	2.42
	20	98.63	100.22	0.2	0.68	100.37	99.69	1.71	0.35
	40	93.19	103.16	0.55	0.17	91.51	103.68	2.5	0.64

**Table 3 foods-10-02200-t003:** PAH4 concentration in herbal medicines analyzed using typical pretreatment method.

Sample	Concentration (µg/kg) ^(1)^				
	BaA	CHR	BbF	BaP	PAH4
Sample 1	N.D. ^(2)^	N.D.	N.D.	N.D.	N.D.
Sample 2	0.07 ± 0.00	0.01 ± 0.00	N.D.	N.D.	0.08
Sample 3	0.07 ± 0.00	0.01 ± 0.00	N.D.	N.D.	0.09
Sample 4	0.32 ± 0.01	0.35 ± 0.01	1.34 ± 0.04	0.2 ± 0.01	2.22
Sample 5	0.07 ± 0.00	0.01 ± 0.00	N.D.	N.D.	0.08
Sample 6	0.39 ± 0.04	0.44 ± 0.06	1.78 ± 0.28	0.26 ± 0.04	2.87
Sample 7	0.12 ± 0.00	0.07 ± 0.01	N.D.	0.01 ± 0.00	0.19
Sample 8	0.19 ± 0.00	0.17 ± 0.01	0.46 ± 0.02	0.07 ± 0.00	0.88
Sample 9	0.10 ± 0.00	0.04 ± 0.00	N.D.	N.D.	0.14
Sample 10	0.30 ± 0.01	0.32 ± 0.01	1.21 ± 0.06	0.18 ± 0.01	2.01
Sample 11	0.28 ± 0.02	0.30 ± 0.02	1.08 ± 0.11	0.16 ± 0.02	1.82
Sample 12	0.08 ± 0.00	0.01 ± 0.00	N.D.	N.D.	0.09
Sample 13	0.07 ± 0.00	0.01 ± 0.00	N.D.	N.D.	0.08
Sample 14	0.07 ± 0.00	0.01 ± 0.00	N.D.	N.D.	0.08
Sample 15	0.29 ± 0.01	0.30 ± 0.02	1.09 ± 0.09	0.16 ± 0.01	1.84
Sample 16	0.07 ± 0.00	0.01 ± 0.00	N.D.	N.D.	0.08
Sample 17	0.31 ± 0.01	0.33 ± 0.01	1.23 ± 0.04	0.18 ± 0.01	2.04
Sample 18	0.30 ± 0.01	0.32 ± 0.01	1.17 ± 0.04	0.17 ± 0.01	1.96
Sample 19	0.07 ± 0.00	0.01 ± 0.00	N.D.	N.D.	0.08
Sample 20	N.D.	N.D.	N.D.	N.D.	N.D.
Sample 21	0.07 ± 0.00	0.01 ± 0.00	N.D.	N.D.	0.08
Sample 22	0.30 ± 0.00	0.32 ± 0.01	1.20 ± 0.02	0.18 ± 0.00	2.00
Sample 23	0.30 ± 0.01	0.31 ± 0.02	1.17 ± 0.10	0.17 ± 0.01	1.95
Sample 24	0.07 ± 0.00	0.01 ± 0.00	N.D.	N.D.	0.08
Sample 25	N.D.	N.D.	N.D.	N.D.	N.D.
Sample 26	N.D.	N.D.	N.D.	N.D.	N.D.
Sample 27	0.07 ± 0.00	0.01 ± 0.00	N.D.	N.D.	0.08
Sample 28	N.D.	N.D.	N.D.	N.D.	N.D.
Sample 29	N.D.	N.D.	N.D.	N.D.	N.D.
Sample 30	0.08 ± 0.00	0.02 ± 0.00	N.D.	N.D.	0.10
Sample 31	0.34 ± 0.00	0.38 ± 0.00	1.48 ± 0.01	0.21 ± 0.00	2.42
Sample 32	0.40 ± 0.00	0.46 ± 0.01	1.84 ± 0.03	0.27 ± 0.00	2.97
Sample 33	0.32 ± 0.04	0.35 ± 0.05	1.31 ± 0.24	0.19 ± 0.03	2.17
Sample 34	0.29 ± 0.03	0.31 ± 0.03	1.14 ± 0.16	0.17 ± 0.02	1.91
Sample 35	0.07 ± 0.00	0.01 ± 0.00	N.D.	N.D.	0.08
Sample 36	0.08 ± 0.00	0.02 ± 0.00	N.D.	N.D.	0.10
Sample 37	0.07 ± 0.00	0.01 ± 0.00	N.D.	N.D.	0.08
Sample 38	0.30 ± 0.00	0.32 ± 0.00	1.19 ± 0.01	0.17 ± 0.00	1.98
Sample 39	0.31 ± 0.02	0.33 ± 0.03	1.25 ± 0.14	0.18 ± 0.02	2.08
Sample 40	0.07 ± 0.00	0.01 ± 0.00	N.D.	N.D.	0.08
Sample 41	0.07 ± 0.00	0.01 ± 0.00	N.D.	N.D.	0.08
Sample 42	0.32 ± 0.01	0.35 ± 0.01	1.32 ± 0.04	0.19 ± 0.00	2.18
Sample 43	N.D.	N.D.	N.D.	N.D.	N.D.
Sample 44	0.36 ± 0.05	0.40 ± 0.07	1.57 ± 0.35	0.23 ± 0.05	2.56
Sample 45	N.D.	N.D.	N.D.	N.D.	N.D.
Sample 46	0.32 ± 0.02	0.34 ± 0.03	1.30 ± 0.13	0.19 ± 0.02	2.15
Sample 47	0.36 ± 0.05	0.40 ± 0.07	1.58 ± 0.33	0.23 ± 0.05	2.57
Sample 48	0.07 ± 0.00	0.01 ± 0.00	N.D.	N.D.	0.08
Sample 49	N.D.	N.D.	N.D.	N.D.	N.D.
Sample 50	0.07 ± 0.00	0.01 ± 0.01	N.D.	N.D.	0.09

^(1)^ Concentration values were expressed with mean ± standard deviation. ^(2)^ Concentration values below LOD were expressed as N.D. (not detected).

**Table 4 foods-10-02200-t004:** PAH4 concentration in herbal medicines analyzed using QuEChERS pretreatment method.

Sample	Concentration (µg/kg) ^(1)^				
	BaA	CHR	BbF	BaP	PAH4
Sample 1	N.D. ^(2)^	N.D.	0.26 ± 0.03	N.D.	0.26
Sample 2	N.D.	0.05 ± 0.00	0.25 ± 0.01	0.04 ± 0.00	0.35
Sample 3	N.D.	0.05 ± 0.00	0.25 ± 0.00	0.04 ± 0.00	0.35
Sample 4	0.47 ± 0.01	0.64 ± 0.01	2.52 ± 0.06	0.38 ± 0.01	4.01
Sample 5	N.D.	0.05 ± 0.00	0.25 ± 0.00	0.04 ± 0.00	0.34
Sample 6	0.61 ± 0.09	0.80 ± 0.10	3.14 ± 0.40	0.47 ± 0.06	5.02
Sample 7	0.02 ± 0.01	0.15 ± 0.01	0.63 ± 0.05	0.10 ± 0.01	0.91
Sample 8	0.17 ± 0.01	0.32 ± 0.01	1.28 ± 0.03	0.19 ± 0.00	1.97
Sample 9	N.D.	0.11 ± 0.00	0.46 ± 0.00	0.07 ± 0.00	0.64
Sample 10	0.42 ± 0.02	0.59 ± 0.02	2.33 ± 0.08	0.35 ± 0.01	3.70
Sample 11	0.38 ± 0.04	0.55 ± 0.04	2.15 ± 0.16	0.32 ± 0.02	3.40
Sample 12	N.D.	0.06 ± 0.00	0.27 ± 0.00	0.05 ± 0.00	0.37
Sample 13	N.D.	0.05 ± 0.00	0.23 ± 0.01	0.04 ± 0.00	0.31
Sample 14	N.D.	0.05 ± 0.00	0.25 ± 0.00	0.04 ± 0.00	0.34
Sample 15	0.38 ± 0.03	0.55 ± 0.03	2.17 ± 0.13	0.33 ± 0.02	3.43
Sample 16	N.D.	0.04 ± 0.00	0.22 ± 0.00	0.04 ± 0.00	0.31
Sample 17	0.43 ± 0.01	0.60 ± 0.01	2.36 ± 0.06	0.35 ± 0.01	3.75
Sample 18	0.41 ± 0.01	0.58 ± 0.01	2.28 ± 0.05	0.34 ± 0.01	3.62
Sample 19	N.D.	0.05 ± 0.00	0.23 ± 0.00	0.04 ± 0.00	0.32
Sample 20	N.D.	N.D.	0.22 ± 0.01	N.D.	0.22
Sample 21	N.D.	0.05 ± 0.00	0.25 ± 0.00	0.04 ± 0.00	0.34
Sample 22	0.42 ± 0.01	0.59 ± 0.01	2.32 ± 0.03	0.35 ± 0.01	3.68
Sample 23	0.41 ± 0.03	0.58 ± 0.03	2.27 ± 0.13	0.34 ± 0.02	3.60
Sample 24	N.D.	0.04 ± 0.00	0.22 ± 0.00	0.04 ± 0.00	0.30
Sample 25	N.D.	N.D.	0.22 ± 0.01	N.D.	0.22
Sample 26	N.D.	N.D.	0.22 ± 0.00	N.D.	0.22
Sample 27	N.D.	0.05 ± 0.00	0.23 ± 0.01	0.04 ± 0.00	0.31
Sample 28	N.D.	N.D.	0.21 ± 0.01	N.D.	0.21
Sample 29	N.D.	N.D.	0.22 ± 0.00	N.D.	0.22
Sample 30	N.D.	0.07 ± 0.00	0.31 ± 0.01	0.05 ± 0.00	0.43
Sample 31	0.51 ± 0.00	0.69 ± 0.00	2.71 ± 0.01	0.41 ± 0.00	4.32
Sample 32	0.63 ± 0.01	0.82 ± 0.01	3.23 ± 0.04	0.48 ± 0.01	5.16
Sample 33	0.46 ± 0.08	0.63 ± 0.09	2.48 ± 0.33	0.37 ± 0.05	3.94
Sample 34	0.40 ± 0.05	0.57 ± 0.06	2.23 ± 0.23	0.34 ± 0.03	3.53
Sample 35	N.D.	0.05 ± 0.00	0.24 ± 0.01	0.04 ± 0.00	0.33
Sample 36	N.D.	0.07 ± 0.01	0.31 ± 0.02	0.05 ± 0.00	0.43
Sample 37	N.D.	0.05 ± 0.00	0.25 ± 0.00	0.04 ± 0.00	0.35
Sample 38	0.41 ± 0.00	0.58 ± 0.00	2.30 ± 0.01	0.35 ± 0.00	3.65
Sample 39	0.44 ± 0.05	0.61 ± 0.05	2.40 ± 0.20	0.36 ± 0.03	3.80
Sample 40	N.D.	0.05 ± 0.00	0.23 ± 0.00	0.04 ± 0.00	0.32
Sample 41	N.D.	0.05 ± 0.00	0.24 ± 0.01	0.04 ± 0.00	0.33
Sample 42	0.46 ± 0.01	0.63 ± 0.01	2.49 ± 0.05	0.37 ± 0.01	3.96
Sample 43	N.D.	N.D.	0.22 ± 0.02	N.D.	0.22
Sample 44	0.54 ± 0.12	0.72 ± 0.13	2.84 ± 0.49	0.42 ± 0.07	4.53
Sample 45	N.D.	N.D.	0.23 ± 0.01	N.D.	0.23
Sample 46	0.45 ± 0.04	0.63 ± 0.05	2.46 ± 0.18	0.37 ± 0.03	3.91
Sample 47	0.54 ± 0.11	0.73 ± 0.12	2.85 ± 0.47	0.43 ± 0.07	4.55
Sample 48	N.D.	0.04 ± 0.01	0.22 ± 0.02	0.04 ± 0.00	0.30
Sample 49	N.D.	N.D.	N.D.	N.D.	N.D.
Sample 50	N.D.	0.05 ± 0.01	0.25 ± 0.04	0.04 ± 0.01	0.35

^(1)^ Concentration values were expressed with mean ± standard deviation. ^(2)^ Concentration values below the LOD were expressed as N.D. (not detected).

**Table 5 foods-10-02200-t005:** TEQ values for PAH4 concentration in herbal medicines analyzed using typical pretreatment method.

Sample	TEQ Value (µg/kg)				
	BaA	CHR	BbF	BaP	PAH4
TEF	0.10	0.01	0.10	1.00	
Sample 1	N.D. ^(1)^	N.D.	N.D.	N.D.	N.D.
Sample 2	7.34 × 10^−3^	1.10 × 10^−4^	N.D.	N.D.	7.45 × 10^−3^
Sample 3	7.37 × 10^−3^	1.14 × 10^−4^	N.D.	N.D.	7.48 × 10^−3^
Sample 4	3.24 × 10^−2^	3.52 × 10^−3^	1.34 × 10^−1^	1.96 × 10^−1^	3.66 × 10^−1^
Sample 5	7.31 × 10^−3^	1.07 × 10^−4^	N.D.	N.D.	7.42 × 10^−3^
Sample 6	3.91 × 10^−2^	4.44 × 10^−3^	1.78 × 10^−1^	2.57 × 10^−1^	4.78 × 10^−1^
Sample 7	1.16 × 10^−2^	6.84 × 10^−4^	N.D.	9.17 × 10^−3^	2.14 × 10^−2^
Sample 8	1.87 × 10^−2^	1.65 × 10^−3^	4.58 × 10^−2^	7.32 × 10^−2^	1.39 × 10^−1^
Sample 9	9.69 × 10^−3^	4.29 × 10^−4^	N.D.	N.D.	1.01 × 10^−1^
Sample 10	3.03 × 10^−2^	3.23 × 10^−3^	1.21 × 10^−1^	1.77 × 10^−1^	3.32 × 10^−1^
Sample 11	2.83 × 10^−2^	2.96 × 10^−3^	1.08 × 10^−1^	1.60 × 10^−1^	2.99 × 10^−1^
Sample 12	7.56 × 10^−3^	1.40 × 10^−4^	N.D.	N.D.	7.70 × 10^−3^
Sample 13	7.08 × 10^−3^	7.54 × 10^−5^	N.D.	N.D.	7.16 × 10^−3^
Sample 14	7.31 × 10^−3^	1.06 × 10^−4^	N.D.	N.D.	7.42 × 10^−3^
Sample 15	2.85 × 10^−2^	2.99 × 10^−3^	1.09 × 10^−1^	1.61 × 10^−1^	3.02 × 10^−1^
Sample 16	7.03 × 10^−3^	6.86 × 10^−5^	N.D.	N.D.	7.10 × 10^−3^
Sample 17	3.06 × 10^−2^	3.28 × 10^−3^	1.23 × 10^−1^	1.80 × 10^−1^	3.37 × 10^−1^
Sample 18	2.98 × 10^−2^	3.16 × 10^−3^	1.17 × 10^−1^	1.72 × 10^−1^	3.23 × 10^−1^
Sample 19	7.16 × 10^−3^	8.50 × 10^−5^	N.D.	N.D.	7.24 × 10^−3^
Sample 20	N.D.	N.D.	N.D.	N.D.	N.D.
Sample 21	7.28 × 10^−3^	1.02 × 10^−4^	N.D.	N.D.	7.38 × 10^−3^
Sample 22	3.02 × 10^−2^	3.22 × 10^−3^	1.20 × 10^−1^	1.76 × 10^−1^	3.30 × 10^−1^
Sample 23	2.96 × 10^−2^	3.14 × 10^−3^	1.17 × 10^−1^	1.71 × 10^−1^	3.21 × 10^−1^
Sample 24	6.99 × 10^−3^	6.21 × 10^−5^	N.D.	N.D.	7.05 × 10^−3^
Sample 25	N.D.	N.D.	N.D.	N.D.	N.D.
Sample 26	N.D.	N.D.	N.D.	N.D.	N.D.
Sample 27	7.08 × 10^−3^	7.51 × 10^−5^	N.D.	N.D.	7.16 × 10^−3^
Sample 28	N.D.	N.D.	N.D.	N.D.	N.D.
Sample 29	N.D.	N.D.	N.D.	N.D.	N.D.
Sample 30	8.01 × 10^−3^	2.02 × 10^−4^	N.D.	N.D.	8.22 × 10^−3^
Sample 31	3.45 × 10^−2^	3.80 × 10^−3^	1.48 × 10^−1^	2.15 × 10^−1^	4.01 × 10^−1^
Sample 32	4.01 × 10^−2^	4.57 × 10^−3^	1.84 × 10^−1^	2.66 × 10^−1^	4.95 × 10^−1^
Sample 33	3.19 × 10^−2^	3.67 × 10^−3^	1.31 × 10^−1^	1.92 × 10^−1^	3.59 × 10^−1^
Sample 34	2.92 × 10^−2^	3.09 × 10^−3^	1.14 × 10^−1^	1.68 × 10^−1^	3.14 × 10^−1^
Sample 35	7.21 × 10^−3^	9.20 × 10^−5^	N.D.	N.D.	7.30 × 10^−3^
Sample 36	8.02 × 10^−3^	2.03 × 10^−4^	N.D.	N.D.	8.23 × 10^−3^
Sample 37	7.34 × 10^−^^3^	1.10 × 10^−4^	N.D.	N.D.	7.45 × 10^−3^
Sample 38	2.99 × 10^−^^2^	3.19 × 10^−3^	1.19 × 10^−1^	1.74 × 10^−1^	3.26 × 10^−1^
Sample 39	3.10 × 10^−^^2^	3.33 × 10^−3^	1.25 × 10^−1^	1.84 × 10^−1^	3.44 × 10^−1^
Sample 40	7.13 × 10^−^^3^	8.15 × 10^−5^	N.D.	N.D.	7.21 × 10^−3^
Sample 41	7.25 × 10^−^^3^	9.80 × 10^−5^	N.D.	N.D.	7.35 × 10^−3^
Sample 42	3.21 × 10^−^^2^	3.48 × 10^−3^	1.32 × 10^−1^	1.93 × 10^−1^	3.61 × 10^−1^
Sample 43	N.D.	N.D.	N.D.	N.D.	N.D.
Sample 44	3.59 × 10^−2^	4.00 × 10^−3^	1.57 × 10^−1^	2.28 × 10^−1^	4.25 × 10^−1^
Sample 45	N.D.	N.D.	N.D.	N.D.	N.D.
Sample 46	3.17 × 10^−2^	3.43 × 10^−3^	1.30 × 10^−1^	1.90 × 10^−1^	3.56 × 10^−1^
Sample 47	3.60 × 10^−2^	4.01 × 10^−3^	1.58 × 10^−1^	2.29 × 10^−1^	4.27 × 10^−1^
Sample 48	7.01 × 10^−3^	6.54 × 10^−5^	N.D.	N.D.	7.08 × 10^−3^
Sample 49	N.D.	N.D.	N.D.	N.D.	N.D.
Sample 50	7.38 × 10^−3^	1.15 × 10^−4^	N.D.	N.D.	7.49 × 10^−3^

^(1)^ Concentration values below the LOD were expressed as N.D. (not detected).

**Table 6 foods-10-02200-t006:** TEQ values for PAH4 concentration in herbal medicines analyzed using QuEChERS pretreatment method.

Sample	TEQ Value (µg/kg)				
	BaA	CHR	BbF	BaP	PAH4
TEF	0.10	0.01	0.10	1.00	
Sample 1	N.D. ^(1)^	N.D.	2.58 × 10^−2^	N.D.	2.58 × 10^−2^
Sample 2	N.D.	5.17 × 10^−4^	2.51 × 10^−2^	4.25 × 10^−2^	6.82 × 10^−2^
Sample 3	N.D.	5.24 × 10^−4^	2.54 × 10^−2^	4.29 × 10^−2^	6.88 × 10^−2^
Sample 4	4.65 × 10^−2^	6.42 × 10^−3^	2.52 × 10^−1^	3.78 × 10^−1^	6.83 × 10^−1^
Sample 5	N.D.	5.12 × 10^−4^	2.49 × 10^−2^	4.22 × 10^−2^	6.76 × 10^−2^
Sample 6	6.10 × 10^−2^	8.01 × 10^−3^	3.14 × 10^−1^	4.68 × 10^−1^	8.51 × 10^−1^
Sample 7	2.18 × 10^−3^	1.51 × 10^−3^	6.33 × 10^−2^	9.89 × 10^−2^	1.66 × 10^−1^
Sample 8	1.74 × 10^−2^	3.19 × 10^−3^	1.28 × 10^−1^	1.94 × 10^−1^	3.43 × 10^−1^
Sample 9	N.D.	1.07 × 10^−3^	4.64 × 10^−2^	7.39 × 10^−2^	1.21 × 10^−1^
Sample 10	4.21 × 10^−2^	5.92 × 10^−3^	2.33 × 10^−1^	3.50 × 10^−1^	6.31 × 10^−1^
Sample 11	3.79 × 10^−2^	5.46 × 10^−3^	2.15 × 10^−1^	3.23 × 10^−1^	5.82 × 10^−1^
Sample 12	N.D.	5.69 × 10^−4^	2.71 × 10^−2^	4.54 × 10^−2^	7.31 × 10^−2^
Sample 13	N.D.	4.57 × 10^−4^	2.28 × 10^−2^	3.91 × 10^−2^	6.24 × 10^−2^
Sample 14	N.D.	5.11 × 10^−4^	2.49 × 10^−2^	4.22 × 10^−2^	6.76 × 10^−2^
Sample 15	3.83 × 10^−2^	5.51 × 10^−3^	2.17 × 10^−1^	3.26 × 10^−1^	5.87 × 10^−1^
Sample 16	N.D.	4.46 × 10^−4^	2.23 × 10^−2^	3.84 × 10^−2^	6.12 × 10^−2^
Sample 17	4.28 × 10^−2^	6.00 × 10^−3^	2.36 × 10^−1^	3.54 × 10^−1^	6.39 × 10^−1^
Sample 18	4.09 × 10^−2^	5.80 × 10^−3^	2.28 × 10^−1^	3.43 × 10^−2^	6.18 × 10^−1^
Sample 19	N.D.	4.74 × 10^−4^	2.34 × 10^−2^	4.01 × 10^−2^	6.40 × 10^−2^
Sample 20	N.D.	N.D.	2.17 × 10^−2^	N.D.	2.17 × 10^−2^
Sample 21	N.D.	5.04 × 10^−4^	2.46 × 10^−2^	4.18 × 10^−2^	6.69 × 10^−2^
Sample 22	4.19 × 10^−2^	5.90 × 10^−3^	2.32 × 10^−1^	3.48 × 10^−1^	6.28 × 10^−1^
Sample 23	4.07 × 10^−2^	5.77 × 10^−3^	2.27 × 10^−1^	3.41 × 10^−1^	6.15 × 10^−1^
Sample 24	N.D.	4.34 × 10^−4^	2.19 × 10^−2^	3.78 × 10^−2^	6.02 × 10^−2^
Sample 25	N.D.	N.D.	2.24 × 10^−2^	N.D.	2.24 × 10^−2^
Sample 26	N.D.	N.D.	2.22 × 10^−2^	N.D.	2.22 × 10^−2^
Sample 27	N.D.	4.57 × 10^−4^	2.28 × 10^−2^	3.91 × 10^−2^	6.23 × 10^−2^
Sample 28	N.D.	N.D.	2.14 × 10^−2^	N.D.	2.14 × 10^−2^
Sample 29	N.D.	N.D.	2.19 × 10^−2^	N.D.	2.19 × 10^−2^
Sample 30	N.D.	6.77 × 10^−4^	3.12 × 10^−2^	5.16 × 10^−2^	8.35 × 10^−2^
Sample 31	5.10 × 10^−2^	6.90 × 10^−3^	2.17 × 10^−1^	4.05 × 10^−1^	7.34 × 10^−1^
Sample 32	6.31 × 10^−2^	8.24 × 10^−3^	3.23 × 10^−1^	4.81 × 10^−1^	8.75 × 10^−1^
Sample 33	4.56 × 10^−2^	6.31 × 10^−3^	2.48 × 10^−1^	3.72 × 10^−1^	6.72 × 10^−1^
Sample 34	3.98 × 10^−2^	5.67 × 10^−3^	2.23 × 10^−1^	3.35 × 10^−1^	6.04 × 10^−1^
Sample 35	N.D.	4.86 × 10^−4^	2.39 × 10^−2^	4.07 × 10^−2^	6.51 × 10^−2^
Sample 36	N.D.	6.79 × 10^−4^	3.13 × 10^−2^	5.17 × 10^−2^	8.37 × 10^−2^
Sample 37	N.D.	5.18 × 10^−4^	2.51 × 10^−2^	4.25 × 10^−2^	6.82 × 10^−2^
Sample 38	4.14 × 10^−2^	5.84 × 10^−3^	2.30 × 10^−1^	3.45 × 10^−1^	6.23 × 10^−1^
Sample 39	4.36 × 10^−2^	6.09 × 10^−3^	2.40 × 10^−1^	3.59 × 10^−1^	6.49 × 10^−1^
Sample 40	N.D.	4.68 × 10^−4^	2.32 × 10^−2^	3.97 × 10^−2^	6.34 × 10^−2^
Sample 41	N.D.	4.97 × 10^−4^	2.43 × 10^−2^	4.13 × 10^−2^	6.61 × 10^−2^
Sample 42	4.59 × 10^−2^	6.34 × 10^−3^	2.49 × 10^−1^	3.74 × 10^−1^	6.75 × 10^−1^
Sample 43	N.D.	N.D.	2.19 × 10^−2^	N.D.	2.19 × 10^−2^
Sample 44	5.40 × 10^−2^	7.24 × 10^−3^	2.84 × 10^−1^	4.25 × 10^−1^	7.70 × 10^−1^
Sample 45	N.D.	N.D.	2.26 × 10^−2^	N.D.	2.26 × 10^−2^
Sample 46	4.52 × 10^−2^	6.26 × 10^−3^	2.46 × 10^−1^	3.69 × 10^−1^	6.67 × 10^−1^
Sample 47	5.43 × 10^−2^	7.27 × 10^−3^	2.85 × 10^−1^	4.26 × 10^−1^	7.73 × 10^−1^
Sample 48	N.D.	4.40 × 10^−4^	2.21 × 10^−2^	3.81 × 10^−2^	6.07 × 10^−2^
Sample 49	N.D.	N.D.	N.D.	N.D.	N.D.
Sample 50	N.D.	5.26 × 10^−4^	2.55 × 10^−2^	4.30 × 10^−2^	6.90 × 10^−2^

^(1)^ Concentration values below the LOD were expressed as N.D. (not detected).

**Table 7 foods-10-02200-t007:** Average DAI and MOE values in herbal medicine products.

Treatment type	DAI of BaP (μg/kg/day)	DAI of PAH4 (μg/kg/day)	MOE
Typical	4.02 × 10^−9^	7.69 × 10^−9^	1.30 × 10^10^
QuEChERS	8.85 × 10^−9^	1.60 × 10^−8^	6.26 × 10^9^

## Data Availability

The datasets generated for this study are available on request to the corresponding author.

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
