# Peer review of "Polycyclic Aromatic Hydrocarbon Risk Assessment and Analytical Methods Using QuEchERS Pretreatment for the Evaluation of Herbal Medicine Ingredients in Korea"

_foods, 2021, doi:10.3390/foods10092200_

Round 1

Reviewer 1 Report

The paper is focused on the method development for different polycyclic aromatic hydrocarbons using QuEChERS method and HPLC. The paper is well written. There are few observations and suggestions:

In the introduction, there are necessary more references related to the methods (GC and HPLC)  for the determination of PAH.

Table 3-6 should be merged.

Table 7 Caption is missing

Author Response

Responses to reviewers’ comments

Manuscript ID: foods-1365613

Title: Polycyclic aromatic hydrocarbon risk assessment and analytical methods using QuEchERS pretreatment for the evaluation of herbal medicine ingredients in Korea

I really appreciate your useful comments and suggestions on our manuscript. We have revised the manuscript accordingly, and detailed changes and explanations are listed below point by point:

Reviewer 1: The paper is focused on the method development for different polycyclic aromatic hydrocarbons using QuEChERS method and HPLC. The paper is well written. There are few observations and suggestions:

In the introduction, there are necessary more references related to the methods (GC and HPLC)  for the determination of PAH.

(Answer) The references related to the GC and HPLC have been added in the manuscript (lines 56-58).

Table 3-6 should be merged.

(Answer) The amount of data is too large to merge all Tables 3-6. Please understand that we keep the table intact for better comparison and readability between data.

Table 7 Caption is missing

(Answer) The caption of Table 7 has been added as the reviewer pointed out.

The manuscript has been resubmitted to your journal. We look forward to your positive response.

Thank you.

Reviewer 2 Report

The main purpose of this study was to determine whether the pretreatment of QuEChERS method could replace the conventional pretreatment method for the determination of PAH4 in herbal medicine ingredients.  The data showed that the QuEChERS pretreatment method coupled with HPLC-FLD could replace the conventional pretreatment method coupled with GC/MS for the determination of PAH4 (BaA, CHR, BbF, and BaP) in herbal medicine ingredients.

  1. Please provide high-resolution graph on Figures 1 and 2 as the labelling of x and y axis was unreadable.
  2. Please check the title of Table 2. It was inconsistent with the table.
  3. As line 274 mentioned, “…, whereas this compound was detected in 21, 41, 49, and 41 samples when the QuEChERS method was used.” However, sample 49 was not detected in Table 4. Please check which one is correct and revise.
  4. What is the meaning in the titles of Table 1 and Table 2: This Comparison of …..
  5. What is the meaning in the description of Table 7: “This is a table. Tables should be placed in the main text near to the first time they are cited.”

Author Response

Responses to reviewers’ comments

Manuscript ID: foods-1365613

Title: Polycyclic aromatic hydrocarbon risk assessment and analytical methods using QuEchERS pretreatment for the evaluation of herbal medicine ingredients in Korea

I really appreciate your useful comments and suggestions on our manuscript. We have revised the manuscript accordingly, and detailed changes and explanations are listed below point by point:

Reviewer 2: The main purpose of this study was to determine whether the pretreatment of QuEChERS method could replace the conventional pretreatment method for the determination of PAH4 in herbal medicine ingredients.  The data showed that the QuEChERS pretreatment method coupled with HPLC-FLD could replace the conventional pretreatment method coupled with GC/MS for the determination of PAH4 (BaA, CHR, BbF, and BaP) in herbal medicine ingredients.

Please provide high-resolution graph on Figures 1 and 2 as the labelling of x and y axis was unreadable.

(Answer) Figure 1 and 2 have been revised as the reviewer pointed out and the original files were also have provided separately.

Please check the title of Table 2. It was inconsistent with the table.

(Answer) The caption of Table 2 has been revised as the reviewer pointed out.

As line 274 mentioned, “…, whereas this compound was detected in 21, 41, 49, and 41 samples when the QuEChERS method was used.” However, sample 49 was not detected in Table 4. Please check which one is correct and revise.

(Answer) The sentence has been revised for clarity (line 274).

What is the meaning in the titles of Table 1 and Table 2: This Comparison of …..

(Answer) The captions of Table 1 and 2 have been revised as the reviewer pointed out.

What is the meaning in the description of Table 7: “This is a table. Tables should be placed in the main text near to the first time they are cited.”

(Answer) The caption of Table 7 has been added and the position of the table has been revised.

The manuscript has been resubmitted to your journal. We look forward to your positive response.

Thank you.
